# Large-Scale Seedling Grow-Out Experiments Do Not Support Seed Transmission of Sweet Potato Leaf Curl Virus in Sweet Potato

**DOI:** 10.3390/plants10010139

**Published:** 2021-01-12

**Authors:** Sharon A. Andreason, Omotola G. Olaniyi, Andrea C. Gilliard, Phillip A. Wadl, Livy H. Williams, D. Michael Jackson, Alvin M. Simmons, Kai-Shu Ling

**Affiliations:** USDA Agricultural Research Service, U.S. Vegetable Laboratory, 2700 Savannah Highway, Charleston, SC 29414, USA; sharon.andreason@usda.gov (S.A.A.); toladosunmu@gmail.com (O.G.O.); andrea.gilliard@usda.gov (A.C.G.); phillip.wadl@usda.gov (P.A.W.); livy.williams@usda.gov (L.H.W.); mike.jackson710@gmail.com (D.M.J.); alvin.simmons@usda.gov (A.M.S.)

**Keywords:** *Geminiviridae*, sweepoviruses, SPLCV, sweet potato, *Ipomoea batatas*, *Bemisia tabaci*, whitefly

## Abstract

Sweet potato leaf curl virus (SPLCV) threatens global sweet potato production. SPLCV is transmitted by *Bemisia tabaci* or via infected vegetative planting materials; however, SPLCV was suggested to be seed transmissible, which is a characteristic that is disputed for geminiviruses. The objective of this study was to revisit the validity of seed transmission of SPLCV in sweet potato. Using large-scale grow-out of sweet potato seedlings from SPLCV-contaminated seeds over 4 consecutive years, approximately 23,034 sweet potato seedlings of 118 genotype entries were evaluated. All seedlings germinating in a greenhouse under insect-proof conditions or in a growth chamber were free of SPLCV; however, a few seedlings grown in an open bench greenhouse lacking insect exclusion tested positive for SPLCV. Inspection of these seedlings revealed that *B. tabaci* had infiltrated the greenhouse. Therefore, transmission experiments were conducted using *B. tabaci* MEAM1, demonstrating successful vector transmission of SPLCV to sweet potato. Additionally, tests on contaminated seed coats and germinating cotyledons demonstrated that SPLCV contaminated a high percentage of seed coats collected from infected maternal plants, but SPLCV was never detected in emerging cotyledons. Based on the results of grow-out experiments, seed coat and cotyledon tests, and vector transmission experiments, we conclude that SPLCV is not seed transmitted in sweet potato.

## 1. Introduction

Sweet potato, *Ipomoea batatas* (L.) Lam., in the morning glory family Convolvulaceae, is the sixth most important food crop worldwide behind rice, wheat, potatoes, maize, and cassava [1]. As a root vegetable high in nutritive value, sweet potato has been relied on during food security crises, serving as a staple crop when other primary crops have failed [2,3]. It also boasts a wide range of cultivation purposes, from subsistence farming to sales in high-profit, health food markets [3]. China is the largest producer of sweet potato, with over 70% of world production; however, sweet potato is grown in all tropical and subtropical regions of the world [4]. Although sweet potato is an indeterminant perennial, its production is often as an annual crop that is initiated by vegetative propagation [5]. As such, planting stocks (e.g., storage roots and plant vines) can accumulate pathogens and serve as inoculum for disease in young crops and as a source of pathogen dissemination. The accumulation of pathogens in vegetatively propagated sweet potato contributes to their decrease in yield over time, resulting in cultivar decline [6].

Sweet potatoes are infected by over 30 different phytopathogenic viruses belonging to nine families [3]. Many of these viruses are vector transmitted, including whitefly-transmitted begomoviruses in the family *Geminiviridae* or aphid-transmitted potyviruses in the family *Potyviridae*. One of the most economically important sweet potato infecting begomovirus (also known as sweepoviruses) species is *Sweet potato leaf curl virus* (SPLCV) [7,8], which is the predominant begomovirus infecting sweet potato in the United States (U.S.) [9]. In a study of the host range of SPLCV, 38 of 45 tested *Ipomoea* spp. were hosts of the virus [10] as well as the hosts for the whitefly *Bemisia tabaci* [11]. In addition to sweet potato, these wild morning glory species could serve as potential natural SPLCV reservoirs. SPLCV and other sweepoviruses have been identified in many major sweet potato growing areas in the world [3].

Sweepoviruses are a group of begomoviruses infecting sweet potato or other plants in the family Convolvulaceae [12]. These sweepoviruses have monopartite circular single-stranded DNA genomes, and some are associated with DNA satellites [13,14,15]. The sweepoviruses are monophyletic, distinct from other old world and new world begomoviruses in phylogenetic analyses [12,16]. Depending on genotype, SPLCV-infected sweet potato plants may develop upward leaf curl symptom on young or newly developed leaves, but symptoms gradually fade in mature plants under field conditions; however, the virus can attain high titer and results in serious yield loss [3,9,17].

The transmission of SPLCV in sweet potato typically occurs via propagation slips from plant beds generated from infected sweet potato. The virus can also be transmitted from infected to healthy plants through vector transmission. Similar to other begomoviruses, SPLCV is transmitted exclusively by the whitefly *B. tabaci* (Hemiptera: Aleyrodidae) in a persistent and circulative manner [18,19]. The whitefly transmission efficiency of SPLCV is demonstrated as low [18,19], often with many individuals needed for transmission [20,21]. Whitefly transmission of SPLCV to different host plant species might also be variable, with a higher efficiency in transmission to morning glory than to sweet potato [19].

Begomoviruses can be seedborne (virus contamination on seeds extracted from an infected mother plant) but are generally not considered to be seed transmitted (vertical transmission of a begomovirus from contaminated seeds to their germinated seedlings). However, seed transmission of SPLCV in sweet potato has been reported [22]. After this first report of begomovirus seed transmissibility, several other reports of seed transmitted begomoviruses were published, including tomato yellow leaf curl virus type strain Israel (TYLCV-IL) in tomato, pepper, and soybean by the same research group [23,24,25]. In addition, several other begomoviruses reported to be seed transmissible are bitter gourd yellow mosaic virus in bitter gourd (*Momordica charantia*) [26], tomato leaf curl New Delhi virus in chayote (*Sechium edule*) [27] and zucchini squash (*Cucurbita pepo*) [28], dolichos yellow mosaic virus in lablab-bean (*Lablab purpureus*) [29], and pepper yellow leaf curl Indonesia virus in pepper (*Capsicum annuum*) in Indonesia [30]. While those reports concluded that seed transmission of various begomoviruses occurred, the studies were typically performed using a very small number of seedlings in grow-out experiments. Kothandaraman et al. [31] reported the detection of mung bean yellow mosaic virus (MYMV) in seed coat, cotyledon, and embryonic axes, but no positive infection in grow-out seedlings, emphasizing the seedborne nature of the begomovirus. No evidence of seed transmission of TYLCV was detected in *Nicotiana benthamiana* [32]. Further investigation into the seedborne nature and possible seed transmission of TYLCV-IL in a large-scale grow-out experiment on seven tomato genotypes by Pérez-Padilla et al. [33] confirmed that the virus can be detected on and in tomato seeds, including the embryo, but that seed-to-seedling virus transmission did not occur, concluding that seed transmission is not a general property of TYLCV-IL. In the present study, our objective was to assess the validity of seed transmission of SPLCV in sweet potato.

## 2. Results

### 2.1. Assessing Seed Transmissibility of SPLCV in Sweet Potato through Seedling Grow-Out in an Open Greenhouse and Real-Time PCR in 2016

In 2016, a total of 3428 seedlings of 20 different sweet potato maternal genotypes were germinated in open bench soil beds in a greenhouse. Fifteen of the 20 maternal genotypes tested positive for SPLCV (Table 1). The average germination rate for each genotype was 79.4%. Six weeks post germination, seedlings in approximately the 4-5 true leaf stage were visually observed for disease-like symptoms. Determination for the presence of SPLCV was based on real-time PCR testing using 70 bulked leaf samples. All bulked samples tested negative for SPLCV (Table 1). The Ct value of one sample (genotype USDA-10-102; Table 1, Figure 1) was close to the threshold Ct value (for calling a positive versus negative test) and therefore was retested. Upon retesting with real-time PCR and conventional PCR, SPLCV was definitively not detected in that sample (Table 1; Figure 1). Further analysis revealed that SPLCV was present in the maternal tissues, including the storage root, flower, and seed of this genotype (USDA-10-102; Figure 1). Thus, in the 2016 grow-out experiment, SPLCV was not detected in 3428 seedlings of 20 genotypes using real-time PCR on germinated sweet potato seedlings (Table 1).

### 2.2. Assessing Seed Transmissibility of SPLCV through Seedling Grow-Out Experiments in Different Growing Conditions in 2017: Under Insect-Proof Netting, in Open Benches in a Greenhouse, and in a Growth Chamber

To evaluate the possible seed transmissibility of SPLCV in sweet potato, we conducted seedling grow-out experiments in three different growing conditions in 2017 (Figure 2). The first experiment was conducted inside insect exclusion cages (1 m^3^ polyvinyl chloride frames with insect-proof netting) in a greenhouse (Figure 2). With a total of 2203 seedlings germinated from seeds of 18 sweet potato genotypes, all seedlings tested in 38 bulked leaf tissue samples by real-time PCR were negative for SPLCV (Table 2). The second experiment was conducted in open bench soil beds without insect-proof netting in a different greenhouse (Figure 2). In this condition, a total of 8744 seedlings were generated from 32 sweet potato genotypes. Although most of the seedlings tested negative, SPLCV was detected in two bulked samples from two different genotypes: ‘Carolina Bunch’ and ‘Macana’ (Table 3). For these sweet potato genotypes, SPLCV was detected in only one of five total bulked samples. Additionally, reactions for the four bulked samples of genotype ‘Hwangmi’ had an average cycle threshold (Ct) value of 31.51 with Ct values of all individual samples (not shown) falling within the borderline range (Ct: 30–35) for positive reaction determination (Table 3). Upon closer inspection, adult *B. tabaci* whiteflies were found to have infiltrated this open bench greenhouse. Therefore, SPLCV infection of those plants was likely due to vector transmission. The third experiment was conducted in an environmental growth chamber inside a headhouse, isolating the seedlings from possible exposure to insect introduction. In a total of 2892 seedlings germinated from 18 genotypes, SPLCV was not detected in any of the 56 bulked samples tested (Table 4).

### 2.3. Evaluation of Seed Coats and Emerging Cotyledons for SPLCV Infection

In response to SPLCV being detected in some seedlings in 2017, we tested the seed coats of germinated seedlings for 10 seeds each of three SPLCV positive maternal genotypes (‘Carolina Bunch’, ‘Regal’, and USDA-09-130; Table 5). After seeds germinated in petri dishes and the dehisced seed coat naturally separated from the seedling, the seed coats and cotyledons were tested for the presence of SPLCV. Most seed coats tested positive for SPLCV, with 90% of ‘Carolina Bunch’, 90% of ‘Regal’, and 30% of USDA-09-130 seed coats testing positive for SPLCV contamination (Table 5). In contrast, SPLCV was not detected in any newly germinated cotyledons (seedlings), demonstrating that the virus was unlikely to have been seed-transmitted.

### 2.4. Assessing Seed Transmissibility of SPLCV in Sweet Potato through Seedling Grow-Out in an Open Greenhouse and Real-Time PCR in 2018

The 2018 seedling grow-out experiment was conducted in open bench soil beds without insect exclusion cages. A total of 4867 seedlings germinated from seeds of 27 sweet potato genotypes (Table 6). Typical leaf curl-like disease symptoms were not observed on any seedlings, and the majority of seedlings tested negative for SPLCV six weeks post germination; however, three of the 76 bulked samples tested positive for SPLCV (genotypes ‘Bayou Belle’, ‘CN 1510-25’, and USDA-11-096; Table 6). As in 2017, SPLCV was detected in only one of multiple tested bulked samples of each genotype, demonstrating few and inconsistent positive results in those genotypes. Additionally, similar to the 2017 open bench grow-out experiment, closer examination found that adult whiteflies had infiltrated this greenhouse, again suggesting the potential whitefly transmission of SPLCV from the environment. 

### 2.5. Assessing Seed Transmissibility of SPLCV in Sweet Potato through Seedling Grow-Out in an Insect-Proof BugDorm^™^ and End-Point PCR in 2019

To eliminate the potential introduction of whiteflies from the outside environment, in the 2019 seedling grow-out experiment, seeds were first germinated in a controlled environment growth chamber, and then seedlings were transferred to insect-proof BugDorm^™^ cages in a greenhouse. Approximately 900 seedlings of three genotypes (‘Bellevue’, ‘Burgundy’, and USDA-08-383) germinated and were tested for presence of SPLCV 8 weeks post germination (Table 7). Seedlings were collected and tested in 13 bulked samples using end-point PCR. SPLCV was not detected in any seedlings in 2019 (Table 7).

### 2.6. Evaluating SPLCV Transmissibility by Vector Transmission Tests

With 4 years of large-scale seedling grow-out experiments, it was difficult to eliminate whiteflies (*B. tabaci* MEAM1; Figure 3A) in a greenhouse environment, even with insecticide applications. Experiments demonstrated that most maternal genotypes were infected with SPLCV and that seeds were contaminated with SPLCV, but the vast majority of grow-out seedlings tested negative for SPLCV (Table 1, Table 2, Table 3, Table 4, Table 5, Table 6 and Table 7). SPLCV was detected in a small number of seedlings; however, positive tests were always associated with seedlings growing in an open bench greenhouse with presence of the SPLCV vector *B. tabaci*. Therefore, we could not rule out the possibility of whitefly transmission of SPLCV. As such, vector transmission tests were performed in 2018 and 2019 to confirm that SPLCV was vector-transmissible and could have been transmitted by *B. tabaci* invading the greenhouse in 2017 and 2018. 

The first transmission experiment was conducted in 2018 on 25 ‘Regal’ sweet potato seedlings after exposure to viruliferous whiteflies. Of 25 plants, SPLCV was detected in one plant, demonstrating 4% transmission efficiency (Table 8). The second transmission experiment was conducted in 2019, with three biological replicates of 60 healthy sweet potato seedlings in separate BugDorm™ cages containing virus-free seedlings of USDA-08-383 or ‘Bellevue’ (Figure 3). After exposure to viruliferous whiteflies, seedlings were maintained 6 weeks for the observation of symptom expression and subsequent evaluation for the presence of SPLCV using end-point PCR. Leaf curl symptoms were observed on one plant exposed to viruliferous whiteflies in 2019 (Figure 3). An initial PCR test of 2019 transmission test plants was performed using bulked samples (three bulked samples corresponding to 20 individuals from each cage), one symptomatic plant, one spiked (S) sample for positive control (leaf tissue of one SPLCV-infected plant bulked with leaf tissue of 19 healthy plants), and controls. This test determined that the one plant showing SPLCV symptoms was indeed positive for SPLCV. Bulked samples with just one positive plant in a bulk with 19 parts of healthy tissue still produced a strong PCR product. For the three replicated transmission tests in 2019, SPLCV was detected in zero, two, and three seedlings of 60 seedlings, demonstrating 0%, 3.3%, and 5% transmission efficiencies, respectively (Table 8). These tests confirmed that SPLCV was transmissible by *B. tabaci*, with an overall average transmission efficiency of 3.1%.

## 3. Discussion

It is important to distinguish between ‘seed-borne’ and ‘seed transmission’ when describing the biological properties of plant viruses. Seed transmission of a virus can have a significant impact not only on disease epidemiology and management but also on international seed trade, whereas the seed-borne nature of a virus is less concerning. In the present study, we detected the presence of SPLCV on sweet potato whole seeds and seed coats (testa) as well as other vegetative tissues through real-time PCR detection, confirming SPLCV as seed-borne (contaminant) on sweet potato [22]. Seed transmission indicates that virus particles present on/in the seed could be transmissible to the offspring, resulting in a new infection in the geminated seedlings. Instead of using the artificial separation of different seed parts (seed coats with part of the endosperm or an internal seed tissue of embryo and endosperm) as in Kim et al. 2015 [22] (which might result in a potential cross-contamination of seed parts when testing by PCR), we used germinated seedlings in petri dishes through the natural separation of embryos from seed coats. SPLCV was detected on the seed coats but not in the embryos (Table 5), suggesting that the virus is not seed transmissible.

More importantly, through large-scale testing of seedlings (23,034 sweet potato seedlings of 118 genotype entries) grown in four consecutive years under different growing conditions using diverse genetic materials generated from a sweet potato breeding program, we do not have evidence to support the previous claim of seed transmission of SPLCV in sweet potato [22]. Under strict insect exclusion conditions or in growth chambers, all grow-out seedlings from SPLCV-contaminated seeds of diverse genotypes were free of SPLCV. While we had a small number of seedlings that tested positive for SPLCV, these positive tests were likely due to accidental whitefly transmission to seedlings germinated in greenhouse open benches. Supporting this possibility, SPLCV was not detected in seedlings germinated under the additional protection of insect-proof nettings inside the greenhouse or in enclosed growth chambers nor were whiteflies observed on seedlings grown in these conditions. Using strictly controlled environmental conditions preventing the introduction of whiteflies, our tests on contaminated seeds and early seedlings (cotyledons) demonstrated positive seed coat contamination with SPLCV but a lack of SPLCV detection in the young seedling, indicating that the virus may be present in or on the seed coat but does not infect the developing seedling. Further experiments using viruliferous whiteflies confirmed the successful transmission of SPLCV to sweet potato seedlings, resulting in typical leaf curl symptoms as well as the presence of SPLCV. Overall, results gathered from tests on over 23,000 sweet potato seeds and seedlings of 118 genotype entries (64 different genotypes from a breeding program) did not support the previous conclusion of SPLCV seed transmission. Our study joins two other reports providing evidence disputing seed transmission of begomoviruses, with no detected seed transmission of TYLCV in *N. benthamiana* and tomato in these previous reports [33,34].

While we demonstrated vector transmission of SPLCV from field-collected infected morning glory, our transmission efficiency was low in our experiments; however, the low transmission efficiency of SPLCV by *B. tabaci* in lab tests is well documented [18,19]. Despite the low lab transmission rate, at 3.1%, this rate exceeds that observed in the grow-out experiments (≈1.4% corresponding to five definitively positive bulk samples of 368 total bulk samples). Furthermore, one of our tests showed that only one positive plant per bulked sample results in a clearly positive PCR reaction. It is probable that the percentage of actual positive plants in our grow-out experiments was significantly lower than ≈1.4% with a low number of positive plants in a bulked sample and that greenhouse-infiltrating adult *B. tabaci* may have introduced SPLCV from the outside field.

SPLCV was first reported in the USA in 1999 [7,8] and was shown to be widespread in the southern USA [9] causing serious yield losses in sweet potato [17]. In addition to sweet potato, many *Ipomoea* spp. were also susceptible to SPLCV infection [10] and served as hosts for the whitefly vector *B. tabaci* [11]. SPLCV and other sweepoviruses are widely distributed in many sweet potato growing regions around the world [3]. Although the efficiency of SPLCV transmission by the whitefly *B. tabaci* is relatively low [18,19], sweet potato is a preferred host. Under greenhouse conditions, whitefly populations can reach significant levels and are often difficult to eliminate without additional physical insect exclusion measures. Even in a highly sophisticated greenhouse, it was not surprising to discover whiteflies infesting unprotected plants and SPLCV infection in some of those test plants. Therefore, care must be taken to exclude potential sources of environmental vector whiteflies in studies on seed transmission of begomoviruses. In the field, whiteflies are efficient vectors of begomoviruses. Although inconclusive [35,36,37], the transovarial transmission of begomoviruses by different cryptic species of *B. tabaci* may contribute to virus dissemination [38,39,40]. In our study, we could not exclude the possible *B. tabaci* transmission of SPLCV to the few seedlings that tested positive for the virus in the open bench greenhouse in 2017 and 2018.

In the previous study on seed transmission of SPLCV [22], conclusions of SPLCV transmission to germinating seedlings were based on the results of a low number of grow-out seedlings (n = 99) of a single sweet potato genotype. Seedlings tested in that study were initially germinated in a greenhouse, thereby allowing the possibility of whitefly presence and potential vector transmission. Based on our lack of evidence for seed transmission of SPLCV and that of TYLCV seed transmission studies [33,34], we suggest that those reports of seed transmission of other begomoviruses [22,23,24,25,26,27,28,29,30] be re-examined. Biological assays through the large-scale cultivation of contaminated seedlings using strict conditions excluding any possible introduction of whiteflies present in the environment (not merely testing seeds or embryos for the presence of the targeted begomovirus) will provide more robust evidence for or against seed transmission. If *B. tabaci* is present, even in low numbers, one cannot rule out the potential for vector transmission of begomoviruses.

In conclusion, our data, derived from large-scale seedling grow-out and testing of 23,034 seedlings of 118 maternal genotype entries, dehisced seed coat and cotyledon tests, and vector transmission tests, do not support seed transmission of SPLCV. These results provide additional evidence disputing the seed transmission of begomoviruses [32,33]. With efficient whitefly transmission of begomoviruses, we recommend re-examination of seed transmission claims for this group of vector-transmitted plant viruses.

## 4. Materials and Methods

### 4.1. Generation of Sweet Potato Seeds in a Breeding Program and Evaluation of Their Virus Status in Maternal Materials

The plant material tested in this study is derived from selected sweet potato germplasm that is used in open pollinated crosses within the USDA, ARS, U.S. Vegetable Laboratory Sweet Potato Breeding Program (Charleston, SC, USA). The breeding program uses recurrent mass selection (Appendix A) using an open pollinated polycross system (15–25 parental clones) that relies on natural populations of various insects for cross-pollination [41]. The parental clones used in the breeding nurseries change each year, as they are based on selections made in the previous evaluation cycle. Due to the obligate outcrossing nature, high level of self-incompatibility, and vegetative reproduction via clonal propagation, an individual is fixed for its suite of traits when the selection is made during the evaluation cycle. The length of time that an individual is grown and maintained vegetatively before being selected as a superior genotype leads to most, if not all, selections having acquired virus(es). The time-consuming process (>3–4 years) before an advanced selection is entered into a breeding nursery as a maternal clone serves as an ideal system for the investigation of seed transmission of viruses, especially SPLCV. The plant material tested in this study is derived from seeds harvested from maternal clones that were included in breeding nurseries. Tissues (storage roots, leaves, or seeds) from each maternal clone were tested for SPLCV infection status by real-time PCR [10].

The maternal lines used to generate seedlings for investigation have been cultivated and maintained over many years in field plots and are expected to be infected with SPLCV via natural vector whitefly (*B. tabaci* MEAM1) transmission. Seeds were harvested from individual maternal lines from open pollinated breeding nurseries each year. After seed harvesting, the storage roots from those maternal materials were tested for the presence of SPLCV. In total, seeds from 64 different sweet potato genotypes with 118 maternal sweet potato genotype entries (representing genotype slips newly cultivated each year) in the sweet potato breeding nurseries were used for seedling grow-out experiments over 4 years (2016–2019). Maternal plants were evaluated for SPLCV infection status in 2016 and 2017. In 2018 and 2019, although tests were not conducted to evaluate SPLCV infection in the maternal lines, the genotype entries originated as planting slips derived from previously infected sweet potato and thereby acquired SPLCV via vegetative propagation. The majority of maternal lines used for seedling grow-out experiments tested positive for SPLCV (Table 1, Table 2, Table 3 and Table 4). To evaluate the virus distribution, different tissue types from an SPLCV-infected genotype (USDA-10-102) were evaluated by real-time PCR and end-point PCR, demonstrating virus presence in all maternal tissue types, including storage root, flower, and seeds (Figure 1).

### 4.2. Large-Scale Seedling Grow-Out Experiments under Different Containment Conditions

In collaboration with the seedling evaluation for the sweet potato breeding program, large-scale seedling grow-out experiments were conducted over 4 years (2016–2019) in different containment conditions in greenhouses and growth chambers for the exclusion of insects, particularly whitefly *B. tabaci*, which is endemic in this area (Charleston, SC, USA). To improve seed germination, harvested sweet potato seeds were scarified for 1 h in sulfuric acid (98%) and rinsed in tap water. Treated seeds were planted in greenhouse soil beds or star plug trays (A.M. Leonard Inc., Piqua, OH, USA) or 6-cell seedling starter trays in growth chambers or greenhouses each year.

Seedling grow-out experiments were conducted annually from 2016 to 2019. In 2016, a total of 3428 seedlings from 20 genotypes germinated in a greenhouse with open bench soil beds without insect exclusion cages in a greenhouse (Figure 2). In 2017, seeds were germinated in three different conditions (greenhouse with insect exclusion cages, greenhouse with open bench, and environmental growth chambers; see Figure 2). The two greenhouses were maintained at 30 °C/24 °C day/night and natural light, and the environmental growth chamber was maintained at 22 °C day/night. In the first experiment, a total of 2203 seedlings from 18 genotypes germinated in a greenhouse under insect-proof netting (Figure 2). In the second experiment, a total of 8744 seedlings from 32 genotypes germinated in a greenhouse with open benches (without insect netting; Figure 2). In the third experiment, a total of 2892 seedlings from 18 genotypes germinated in an environmental growth chamber (Figure 2). In 2018, a total of 4867 seedlings from 27 genotypes germinated in a greenhouse with open bench (Figure 2). In 2019, approximately 900 seeds from three genotypes were initially germinated in a growth chamber and then transferred to insect proof BugDorm™ cages (W 47.5 × D 47.5 × H 93.0 cm; MegaView Science Co., Ltd., Taichung City, Taiwan) (similar to Figure 3) and maintained in a greenhouse. In total, approximately 23,034 seedlings from 118 genotype entries were evaluated, with seedlings observed for symptom development and tested for SPLCV using real-time PCR or end-point PCR.

### 4.3. DNA Isolation from Sweet Potato Tissues

DNA was isolated from harvested sweet potato tissues (roots, seed coats, cotyledons, flowers, and seedling leaves) using the DNeasy Plant Mini Kit following the manufacturer’s instructions (Qiagen, Hilden, Germany). Specifically, DNA was isolated from seedling leaf tissues by taking cuttings of approximately 1 cm^2^ of young sweet potato leaves. Leaf cuttings were either bulked or individually macerated in 1.5 mL of AP1 buffer (Qiagen, Germantown, MD, USA) in an extraction bag (Bioreba, Reinach, Switzerland) using a Homex 6 homogenizer (Bioreba, Reinach, Switzerland). Cuttings of storage roots were similarly transferred to extraction bags for maceration in 1.5 mL of AP1 buffer followed by extraction. Seed coats and cotyledons were placed individually into 1.5 mL microcentrifuge tubes and homogenized with micropestles in AP1 buffer. Tissue homogenates (400 µL) were transferred to a microcentrifuge, and DNA was isolated following the DNeasy Plant Mini Kit protocol (Qiagen, Germantown, MD, USA). The resulting concentrations of sweet potato extracts were measured using a Qubit Fluorometer.

### 4.4. SPLCV Testing by Real-Time PCR and End-Point PCR

Quantitative real-time PCR (qPCR) and end-point PCR were performed to detect SPLCV in plant tissues. qPCR tests were conducted using SPLCV-specific primers (Appendix A) described by Ling et al. [10] as follows: TaqMan probe 5′FAM-TACACTGGGAAGCTGTCCCAATTGCT-TAMRA, with forward primer 5′GAGACAGCTATCGTGCC and reverse primer 5′GAAACCGGGACATAGCTTCG, and Takara One Step PrimeScript Real-time PCR Kit following the manufacturer’s instructions (Clontech, Mountain View, CA, USA). The qPCR reaction was carried out on a Stratagene MX3000P Real-Time PCR machine (Agilent Technologies, Santa Clara, CA, USA), under the following conditions: 95 °C for 10 min, followed by 40 cycles of 95 °C for 30 s, 55 °C for 1 min, and 72 °C for 1 min. A Ct value of 30 was determined to be the threshold for a definitively positive SPLCV detection test. Reactions resulting in Ct values between 30 and 35 were considered borderline, and the Ct value of the healthy tissue control in the same test was considered to determine SPLCV detection. Reactions with Ct values above 35 were considered negative for SPLCV.

End-point PCR tests were performed using SPLCV-specific primers SPLCV CP-F (5′-AAG AAA TAC GAG CCA GGA AC) and SPLCV CP-R1 (5′-TAT TAA TTA TTG TGC GAA TCA) (Appendix A). Reactions of 20 μL were comprised of 10 μL GoTaq^®^ Green MasterMix (Promega, Madison, WI, USA), 0.5 ul of forward and reverse primers (20 μM), 1 μL of template DNA, and 8 μL of nuclease-free water. Thermal cycling was programed on an Eppendorf Mastercycler Nexus™ as follows: 94 °C for 2 min, 38 cycles of 94 °C for 30 s, 55 °C for 1 min, 72 °C for 1 m, and a final extension step at 72 °C for 10 min. Positive controls (SPLCV-infected sweet potato), negative controls (healthy sweet potato), and non-template controls (nuclease-free water) were included in each test. Product sizes were confirmed by electrophoresis on a 1.5% agarose gel in Tris-acetate-EDTA (TAE) buffer stained with SYBR^®^ Safe DNA Gel Stain (Invitrogen, Carlsbad, CA, USA) using a Lambda Hind III™ 1 Kb DNA Ladder (Invitrogen, Carlsbad, CA, USA) and PCR Marker (Promega. Fitchburg, WI, USA).

### 4.5. SPLCV Seed Coat Testing

To eliminate the potential impact of vector transmission of SPLCV in determining seed transmission, dehisced seed coats from germinated seedlings and newly germinated cotyledons were tested for SPLCV (Appendix A). Ten non-scarified seeds from each of three SPLCV positive maternal genotypes (‘Carolina Bunch’, ‘Regal’, and USDA-09-130) were germinated on sterilized moistened filter paper in individual glass petri dishes. Petri dishes were sealed with parafilm and placed in the dark at room temperature and checked daily until germination. Upon germination, the seedlings shed the seed coats, and both the seed coat and new seedling (cotyledon stage) were aseptically transferred to separate microcentrifuge tubes for DNA isolation and SPLCV testing as previously described.

### 4.6. SPLCV Transmission by B. tabaci

Vector transmission assays were performed to confirm whitefly transmissibility of SPLCV. Adult *B. tabaci* typed as MEAM1 (Figure 3) using the PCR primers and protocols of Andreason et al. [42] for a previous study [43] were collected from a colony reared on collard (*Brassica oleracea* var. *viridis*) maintained in a greenhouse at environmental conditions. Whiteflies were transferred to cuttings of SPLCV-positive morning glory (*Ipomoea setosa*) and given an acquisition access period (AAP) of 72 h in a laboratory maintained at 72 ± 2 °C. After the AAP, the viruliferous whiteflies were transferred to 25 healthy sweet potato seedlings (genotype ‘Regal’) in 2018 for feeding and inoculation until testing. Tests for SPLCV transmission were performed by qPCR at 30 days and 60 days post whitefly introduction as described above. In 2019, viruliferous whiteflies were transferred to three insect-proof BugDorm™ cages each containing 60 healthy sweet potato plants (genotypes ‘Bellevue’ and USDA-08-383; Figure 3) in a greenhouse maintained at 72 ± 2 °C for feeding and inoculation until testing. Tests for SPLCV transmission were performed by end-point PCR 6 weeks after introduction of viruliferous whiteflies as described above.

## Figures and Tables

**Figure 1 plants-10-00139-f001:**
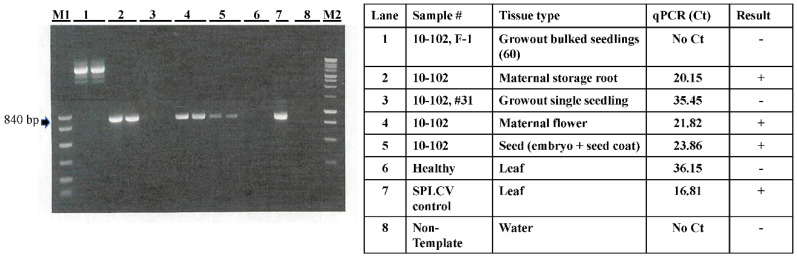
Evaluating the distribution of sweet potato leaf curl virus (SPLCV) on different tissue types of an infected sweet potato using PCR and real-time PCR. The image (left panel) shows gel electrophoresis analysis of products amplified using end-point PCR targeting the coat protein of SPLCV in various sweet potato tissue types, as specified in the table. The table (right panel) presents cycle threshold (Ct) values using real-time PCR testing of the same DNA preparations from various tissues.

**Figure 2 plants-10-00139-f002:**
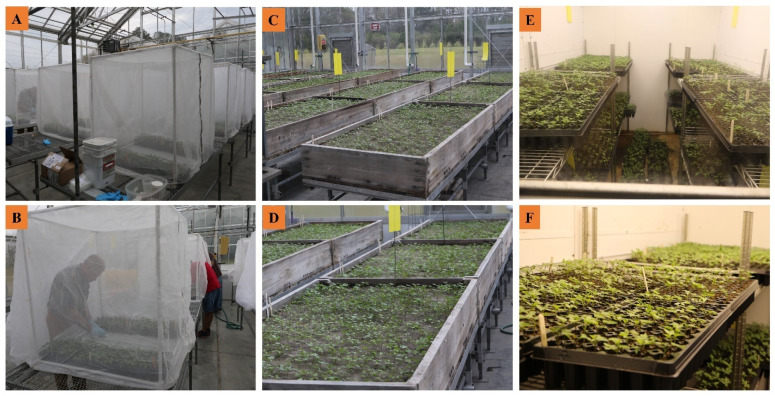
Locations of seedling grow-out experiments. (**A**,**B**) Seedlings grown in insect-proof netting cages in greenhouse #1. (**C**,**D**) Seedlings growing in open bench soil beds in greenhouse #2. (**E**,**F**) Seedlings growing in a growth chamber inside a headhouse.

**Figure 3 plants-10-00139-f003:**
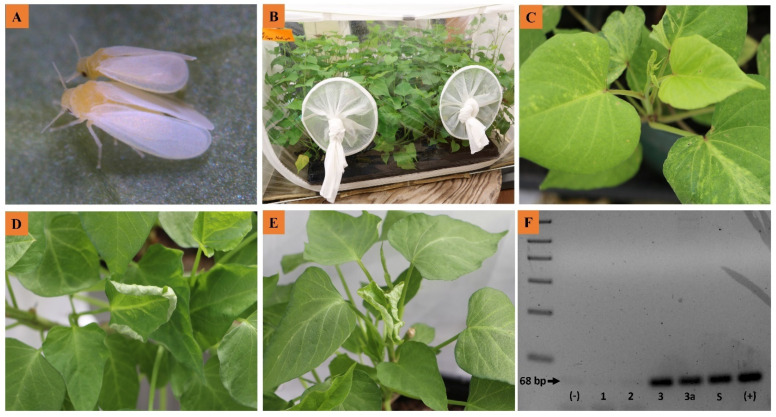
Whitefly transmission of sweet potato leaf curl virus (SPLCV) on sweet potato seedlings. (**A**) *Bemisia tabaci* MEAM1 male and female whiteflies [34]. (**B**) Caged healthy sweet potato seedlings upon introduction of viruliferous whiteflies. (**C**) Asymptomatic sweet potato seedlings. (**D**,**E**) Symptomatic sweet potato seedlings with characteristic upward curling of young leaves. (**F**) End-point PCR product (68 bp) detected on an agarose gel depicting SPLCV positive for a bulked sample (lane 3), the symptomatic plant pictured in D and E (lane 3a), spiked sample with one SPLCV positive plant bulked with 19 healthy ones (lane S), and a positive control (lane +). Two other bulked samples (lanes 1 and 2) and a healthy control (lane −) were negative for SPLCV.

**Table 1 plants-10-00139-t001:** Evaluation of seed transmission of sweet potato leaf curl virus (SPLCV) through grow-out and real-time PCR testing of seedlings grown in a greenhouse in 2016.

Sample No.	Maternal Genotype ^a^	Maternal SPLCV Status (Ct) ^b^	Bulked Tissue DNA Samples ^c^	Tissue Type	Total No. Seedlings Tested	Mean qPCR Ct
1	DM04-226NC	18.92	1	Bulked Leaves	55	No Ct
2	USDA-10-102	19.84	1	Bulked Leaves	60	No Ct ^d^
3	USDA-10-060	18.44	1	Bulked Leaves	50	No Ct
4	USDA-10-188	17.38	4	Bulked Leaves	181	No Ct
5	USDA-10-090	25.58	4	Bulked Leaves	185	No Ct
6	USDA-07-189	16.84	4	Bulked Leaves	184	No Ct
7	USDA-07-135	38.58	4	Bulked Leaves	202	No Ct
8	USDA-11-069	No Ct	4	Bulked Leaves	188	No Ct
9	‘Ruddy’	16.78	4	Bulked Leaves	180	No Ct
10	NC04-531	18.82	4	Bulked Leaves	175	No Ct
11	USDA-09-083	34.24	4	Bulked Leaves	205	No Ct
12	USDA-11-094	35.75	4	Bulked Leaves	195	No Ct
13	USDA-08-779NC	34.74	4	Bulked Leaves	214	No Ct
14	USDA-11-096	31.25	4	Bulked Leaves	182	No Ct
15	USDA-10-087	24.82	4	Bulked Leaves	177	No Ct
16	USDA-10-073NC	30.29	4	Bulked Leaves	201	No Ct
17	USDA-09-050	36.06	4	Bulked Leaves	173	No Ct
18	‘Charleston Scarlet’	18.47	3	Bulked Leaves	205	No Ct
19	USDA-10-100	No Ct	4	Bulked Leaves	193	No Ct
20	USDA-11-032	16.66	4	Bulked Leaves	223	No Ct
	**Total**		70		3428	No Ct
	Healthy control	No Ct		Leaf	1	No Ct
	SPLCV control	18.13		Leaf	1	15.80
	Non-template control	No Ct				No Ct

^a^ Storage roots of mother plants were tested for the presence of SPLCV using real-time PCR. Seeds were collected from plants evaluated in polycross breeding in a field in Charleston, SC. ^b^ Threshold value for a positive maternal sample in real-time PCR was Ct: 35.00. ^c^ Number of DNA samples prepared from bulked seedling leaf disks and tested for the presence of SPLCV using real-time PCR. ^d^ The first test of bulked leaf sample genotype USDA-10-102 had a Ct value close to the threshold positive value. Therefore, this sample was retested, resulting in a negative test.

**Table 2 plants-10-00139-t002:** Evaluation of seed transmission of sweet potato leaf curl virus (SPLCV) through grow-out and real-time PCR testing of seedlings grown inside insect-proof netting in a greenhouse in 2017.

Sample No.	Maternal Genotype ^a^	Maternal SPLCV Status (Ct) ^b^	Bulked Tissue DNA Samples ^c^	Tissue Type	Total No. Seedlings Tested	Mean qPCR Ct
1	USDA-02-415	25.33	2	Bulked leaves	91	No Ct
2	USDA-04-412	22.18	2	Bulked leaves	101	No Ct
3	USDA-08-383	22.52	3	Bulked leaves	155	No Ct
4	USDA-10-100	22.96	2	Bulked leaves	113	No Ct
5	USDA-11-096	32.99	2	Bulked leaves	104	No Ct
6	USDA-95-145	27.52	1	Bulked leaves	48	No Ct
7	‘Bellevue’	33.24	2	Bulked leaves	118	No Ct
8	‘Burgundy’	22.73	2	Bulked leaves	108	No Ct
9	‘Carolina Bunch’	22.50	2	Bulked leaves	105	No Ct
10	‘Excel’	35.05	4	Bulked leaves	212	No Ct
11	CN 1510-25	29.64	2	Bulked leaves	124	No Ct
12	‘Hwangmi’	22.80	2	Bulked leaves	101	No Ct
13	W-154	22.80	2	Bulked leaves	128	No Ct
14	‘Harris Purple’	18.80	2	Bulked leaves	98	No Ct
15	‘Liberty’	22.98	2	Bulked leaves	109	No Ct
16	‘O’Henry’	33.22	1	Bulked leaves	54	No Ct
17	‘Regal’	23.63	4	Bulked leaves	196	No Ct
18	‘Ruddy’	30.73	4	Bulked leaves	238	No Ct
	**Total**		38	Bulked leaves	2203	No Ct
	Healthy control	No Ct		Leaf	1	No Ct
	SPLCV control	18.13		Leaf	1	18.59
	Non-template control	No Ct			1	No Ct

^a^ Storage roots of mother plants were tested for the presence of SPLCV using real-time PCR. Seeds were collected from plants evaluated in polycross breeding in a field in Charleston, SC. ^b^ Threshold value for a positive maternal sample in real-time PCR was Ct: 35.00. ^c^ Number of DNA samples prepared from bulked seedling leaf disks and tested for the presence of SPLCV using real-time PCR.

**Table 3 plants-10-00139-t003:** Evaluation of seed transmission of sweet potato leaf curl virus (SPLCV) through grow-out and real-time PCR testing of seedlings grown in a greenhouse in 2017.

Sample No.	Maternal Genotype ^a^	Maternal SPLCV Status (Ct) ^b^	Bulked Tissue DNA Samples ^c^	Tissue Type	Total No. Seedlings Tested	Mean qPCR Ct
1	USDA-00-099	15.78	2	Bulked leaves	71	No Ct
2	USDA-02-015	17.93	1	Bulked leaves	133	No Ct
3	USDA-02-415	25.33	2	Bulked leaves	151	No Ct
4	USDA-04-284	No Ct	1	Bulked leaves	72	No Ct
5	USDA-04-412	22.18	2	Bulked leaves	167	No Ct
6	USDA-05-097	18.61	5	Bulked leaves	378	No Ct
7	USDA-08-383	22.52	3	Bulked leaves	196	No Ct
8	USDA-09-130	18.70	2	Bulked leaves	198	No Ct
9	USDA-10-015	22.97	7	Bulked leaves	528	No Ct
10	USDA-10-174	17.92	1	Bulked leaves	83	No Ct
11	USDA-11-096	32.99	4	Bulked leaves	382	No Ct
12	USDA-95-145	27.52	5	Bulked leaves	174	No Ct
13	USDA-97-095	18.21	1	Bulked leaves	28	No Ct
14	‘Bellevue’	33.24	5	Bulked leaves	367	No Ct
15	‘Burgundy’	22.73	4	Bulked leaves	340	No Ct
16	‘Carolina Bunch’	22.50	4	Bulked leaves	323	No Ct
			1	Bulked leaves	68	19.20
17	‘Charleston Scarlet’	No Ct	2	Bulked leaves	146	No Ct
18	‘Evangeline’	17.84	1	Bulked leaves	91	No Ct
19	‘Excel’	35.05	8	Bulked leaves	699	No Ct
20	‘Macana’	24.32	4	Bulked leaves	314	No Ct
			1	Bulked leaves	77	17.95
21	CN1510-25	29.64	5	Bulked leaves	440	No Ct
22	‘Georgia Jet’	No Ct	2	Bulked leaves	183	No Ct
23	‘Hwangmi’	22.80	4	Bulked leaves	326	31.51
24	AC 83.4-2	No Ct	2	Bulked leaves	179	No Ct
25	W-154	22.80	4	Bulked leaves	354	No Ct
26	Harris Purple	18.80	2	Bulked leaves	182	No Ct
27	‘Liberty’	22.98	3	Bulked leaves	178	No Ct
28	‘O’Henry’	33.22	3	Bulked leaves	197	No Ct
29	‘Regal’	23.63	8	Bulked leaves	641	No Ct
30	‘Ruddy’	21.58	10	Bulked leaves	750	No Ct
31	‘Sumor’	14.46	3	Bulked leaves	180	No Ct
32	W-390	21.59	3	Bulked leaves	148	No Ct
	**Total**		115	Bulked leaves	8744	No Ct
	Healthy Control	No Ct		Leaf	1	37.24
	SPLCV Control	18.13		Leaf	1	23.81
	Non-template control	No Ct				No Ct

^a^ Storage roots of mother plants were tested for the presence of SPLCV using real-time PCR. Seeds were collected from plants evaluated in polycross breeding in a field in Charleston, SC. ^b^ Threshold value for a positive maternal sample in real-time PCR was Ct: 35.00. ^c^ Number of DNA samples prepared from bulked seedling leaf disks and tested for the presence of SPLCV using real-time PCR.

**Table 4 plants-10-00139-t004:** Evaluation of seed transmission of sweet potato leaf curl virus (SPLCV) through grow-out and real-time PCR testing of seedlings grown in a growth chamber in 2017.

Sample No.	Maternal Genotype ^a^	Maternal SPLCV Status (Ct) ^b^	Bulked Tissue DNA Samples ^c^	Tissue Type	Total No. Seedlings Tested	Mean qPCR Ct
1	USDA-02-415	25.33	2	Bulked leaves	92	No Ct
2	USDA-04-412	22.18	2	Bulked leaves	81	No Ct
3	USDA-08-383	22.52	3	Bulked leaves	156	No Ct
4	USDA-10-100	22.96	3	Bulked leaves	162	No Ct
5	USDA-11-096	32.99	3	Bulked leaves	185	No Ct
6	USDA-95-145	27.52	1	Bulked leaves	36	No Ct
7	‘Bellevue’	33.24	3	Bulked leaves	172	No Ct
8	‘Burgundy’	31.23	3	Bulked leaves	148	No Ct
9	‘Carolina Bunch’	22.50	3	Bulked leaves	166	No Ct
10	‘Excel’	35.05	6	Bulked leaves	323	No Ct
11	CN1510-25	29.64	3	Bulked leaves	162	No Ct
12	‘Hwangmi’	22.80	3	Bulked leaves	134	No Ct
13	W-154	22.80	3	Bulked leaves	171	No Ct
14	Harris Purple	18.80	2	Bulked leaves	82	No Ct
15	‘Liberty’	22.98	3	Bulked leaves	147	No Ct
16	‘O’Henry’	33.22	1	Bulked leaves	48	No Ct
17	‘Regal’	23.63	6	Bulked leaves	285	No Ct
18	‘Ruddy’	21.58	6	Bulked leaves	342	No Ct
	**Total**		56	Bulked leaves	2892	No Ct
	Healthy Control	No Ct		Leaf	1	No Ct
	SPLCV Control	18.13		Leaf	1	23.81
	Non-template control	No Ct				No Ct

^a^ Storage roots of mother plants were tested for the presence of SPLCV using real-time PCR. Seeds were collected from plants evaluated in polycross breeding in a field in Charleston, SC. ^b^ Threshold value for a positive maternal sample in real-time PCR was Ct: 35.00. ^c^ Number of DNA samples prepared from bulked seedling leaf disks and tested for the presence of SPLCV using real-time PCR.

**Table 5 plants-10-00139-t005:** Results of real-time PCR tests on seed coats and germinating cotyledons for three sweet potato leaf curl virus (SPLCV) positive maternal sweet potato genotypes.

	‘Carolina Bunch’		‘Regal’		USDA-09-130	
Sample No.	Seed Coat Mean qPCR Ct ^a^	Cotyledon Mean qPCR Ct	Seed Coat Mean qPCR Ct	Cotyledon Mean qPCR Ct	Seed Coat Mean qPCR Ct	Cotyledon Mean qPCR Ct
1	31.08	No Ct	32.49	No Ct	No Ct	No Ct
2	28.87	No Ct	30.47	No Ct	31.65	No Ct
3	29.58	39.85	31.95	No Ct	No Ct	No Ct
4	26.38	38.55	32.90	No Ct	No Ct	37.79
5	29.32	No Ct	32.01	No Ct	No Ct	No Ct
6	25.99	No Ct	No Ct	No Ct	No Ct	39.18
7	26.83	37.89	31.10	No Ct	38.64	No Ct
8	No Ct	No Ct	30.97	No Ct	30.65	No Ct
9	29.28	No Ct	31.61	No Ct	32.86	No Ct
10	34.89	39.38	34.86	36.86	No Ct	No Ct
**Percent Positive Tests**	90%	0%	90%	0%	30%	0%

^a^ In this real-time PCR test, the healthy sweet potato tissue control had a Ct: 36.61, the SPLCV positive control had a mean Ct: 18.74, and a non-template control had No Ct. The threshold value for positive samples was Ct: 35.00.

**Table 6 plants-10-00139-t006:** Evaluation of seed transmission of sweet potato leaf curl virus (SPLCV) through grow-out and real-time PCR testing of seedlings grown in a greenhouse in 2018.

Sample #	Maternal Genotype	Bulked Tissue DNA Samples ^a^	Tissue Type	Total No. Seedlings Tested	Realtime PCR Ct
1	‘Bayou Belle’	1	Bulked leaves	97	26.31
		3	Bulked leaves	200	No Ct
2	‘Beauregard’	1	Bulked leaves	138	No Ct
3	‘Burgundy’	9	Bulked leaves	829	No Ct
4	‘Charleston Scarlet’	5	Bulked leaves	411	No Ct
5	CN 1108-13	1	Bulked leaves	102	No Ct
6	CN 1510-25	3	Bulked leaves	208	No Ct
		1	Bulked leaves	82	26.76
7	‘Evangeline’	1	Bulked leaves	23	No Ct
8	‘Excel’	3	Bulked leaves	138	No Ct
9	‘Hwangmi’	5	Bulked leaves	298	No Ct
10	‘Orleans’	6	Bulked leaves	314	No Ct
11	‘Regal’	3	Bulked leaves	201	No Ct
12	‘Ruddy’	3	Bulked leaves	237	No Ct
13	USDA-02-015	1	Bulked leaves	19	No Ct
14	USDA-02-126	2	Bulked leaves	173	No Ct
15	USDA-04-136	1	Bulked leaves	48	No Ct
16	USDA-04-791	1	Bulked leaves	17	No Ct
17	USDA-05-097	4	Bulked leaves	156	No Ct
18	USDA-06-001	2	Bulked leaves	160	No Ct
19	USDA-07-135	1	Bulked leaves	83	No Ct
20	USDA-07-182	2	Bulked leaves	84	No Ct
21	USDA-10-126	2	Bulked leaves	87	No Ct
22	USDA-11-028	3	Bulked leaves	143	No Ct
23	USDA-11-096	1	Bulked leaves	97	23.78
		5	Bulked leaves	201	No Ct
24	USDA-16-154	1	Bulked leaves	13	No Ct
25	USDA-16-167	1	Bulked leaves	70	No Ct
26	W-154	3	Bulked leaves	226	No Ct
27	W-259	1	Bulked leaves	12	No Ct
	**Total**	76		4867	No Ct
	Healthy Control			1	No Ct
	SPLCV Control			1	18.57
	Non-template control				No Ct

^a^ Number of DNA samples prepared from bulked seedling leaf disks and tested for the presence of SPLCV using real-time PCR.

**Table 7 plants-10-00139-t007:** Evaluation of seed transmission of sweet potato leaf curl virus (SPLCV) through grow-out and end-point PCR testing of seedlings grown in insect-proof BugDorm™ cages in a greenhouse in 2019.

Maternal Genotype	Bulked Tissue DNA Samples ^a^	Tissue Type	No. Seedlings Tested ^b^	End-Point PCR Detection ^c^
USDA-08-383	4	Bulked leaves	300	(−)
‘Bellevue’	5	Bulked leaves	300	(−)
‘Burgundy’	4	Bulked leaves	300	(−)
**Total**	13		900	(−)
Healthy control		Leaf		(−)
SPLCV positive control		Leaf		(+)
Non-template control				(−)

^a^ Number of DNA samples prepared from bulked seedling leaf disks and tested for the presence of SPLCV using end-point PCR. ^b^ Approximate number of DNA samples prepared from bulked seedling leaf disks and tested for the presence of SPLCV using end-point PCR. ^c^ End-point PCR detection with (+) or without (−) the expected size of product when evaluated on an agarose gel.

**Table 8 plants-10-00139-t008:** Efficiency of *Bemisia tabaci* MEAM1 vector transmission of sweet potato leaf curl virus (SPLCV) to sweet potato seedlings.

	Sweet Potato Genotype	Total No. Seedlings	Positive SPLCV Detection (PCR)	Transmission Rate (%)
**2018 Experiment 1**	‘Regal’	25	1	4.0
**2019 Experiment 2**	USDA-08-383 (cage 1)	60	0	0
	USDA-08-383 (cage 2)	60	2	3.3
	‘Bellevue’	60	3	5.0
**Total**		205	6	3.1 ^a^

^a^ Average transmission rate calculated from experimental replicates.

## Data Availability

The data presented in this study are available in supplementary materials.

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
