# Peer review of "Large-Scale Seedling Grow-Out Experiments Do Not Support Seed Transmission of Sweet Potato Leaf Curl Virus in Sweet Potato"

_plants, 2021, doi:10.3390/plants10010139_

Round 1

Reviewer 1 Report

This is an interesting article that certainly represents the outcome of a significant investment in time and resources. The authors provide strong support for the lack of seed transmission of SPLCV.  

The author’s decision to include the results of experiments that were compromised as a result of whiteflies gaining access to the plant material - and then attributing the subsequent infection(s) to a vector which had supposedly been excluded – after the fact, is unusual. Generally (in my experience) in a situation of that type – the results would be considered invalid (discarded) and the experiment repeated.

Still, the experiments that were not compromised did support the lack of transmission of SPLCV.

The authors statement in line 313 seems premature.

There are some minor grammatical errors (i.e. line 97 – no verb).

Author Response

Reviewer 1:

This is an interesting article that certainly represents the outcome of a significant investment in time and resources. The authors provide strong support for the lack of seed transmission of SPLCV.  

The author’s decision to include the results of experiments that were compromised as a result of whiteflies gaining access to the plant material - and then attributing the subsequent infection(s) to a vector which had supposedly been excluded – after the fact, is unusual. Generally (in my experience) in a situation of that type – the results would be considered invalid (discarded) and the experiment repeated.

Still, the experiments that were not compromised did support the lack of transmission of SPLCV.

Response:  We appreciate the reviewer’s positive comments. We considered leaving out some parts of the grow-out experiments on open beaches in a greenhouse that were compromised as the reviewer suggested, but we decided to keep those data in to offer a whole picture on the experiment as well as to demonstrate the challenges in conducting such experiments on seed transmission where potential efficient incidental whitefly transmission could occur. 

The authors statement in line 313 seems premature.

Response: The statement in line 313 was edited to clarify that, in addition to previous studies on TYLCV (references 32 and 33), our results provide additional evidence disputing seed transmission of begomoviruses.

There are some minor grammatical errors (i.e. line 97 – no verb).

Response: Thank you for pointing out the error here. After closer examination, we realized one of the citations is not proper cited and was removed.  Thus, this sentence has been modified as following: “Kothandaraman et al. [31] reported the detection of mung bean yellow mosaic virus (MYMV) in seed coat, cotyledon, and embryonic axes, but no positive infection in grow-out seedlings, emphasizing the seedborne nature of the begomovirus. All subsequent citation numbers have been adjusted accordingly.     

Reviewer 2 Report

In the submitted manuscript “Large Scale Seedling Grow-out Experiments Do Not 2 Support Seed Transmission of Sweet Potato Leaf 3 Curl Virus in Sweetpotato” by Loebler et al. attempted to determinewhether Sweet potato leaf curl virus(SPLCV) is transmitted via seeds using a large scale grow-out of 23,034 sweetpotato seedlings of 118 genotype accessions under different growing conditions from SPLCV-infected seeds in multiple consecutive years. Although the manuscript presents some evidence for supporting the argument on seed transmission of SPLCV in Sweetpotato, the claim that SPLCV is not seed transmitted in sweetpotato that have not been adequately addressed in the manuscript.

Major concern:

To determine whether SPLCV is transmitted via seeds in Sweetpotato, the published literature (Kim et al., 2015) firstly indicates that SPLCV can be transmitted via seeds in sweet potato plants with up to 15% transmission level, all the sequence of a spanning region in the SPLCV genome from leaf, petal, floral tissues, seeds and seedlings were identical in sequence to that in the Korean isolate of SPLCV. Most importantly, SPLCV was also detected in whole dry seeds, seed coats with part of the endosperm, and internal seed tissue consisting of embryo and endosperm, and the accumulation of SPLCV level was detected by PCR and Southern blot hybridization in developing leaves. Therefore, the authors need to present some comparative evidence to support their argument again the previous claim in the manuscript.

Author Response

Reviewer 2:

In the submitted manuscript “Large Scale Seedling Grow-out Experiments Do Not 2 Support Seed Transmission of Sweet Potato Leaf 3 Curl Virus in Sweetpotato” by Loebler et al. attempted to determinewhether Sweet potato leaf curl virus(SPLCV) is transmitted via seeds using a large scale grow-out of 23,034 sweetpotato seedlings of 118 genotype accessions under different growing conditions from SPLCV-infected seeds in multiple consecutive years. Although the manuscript presents some evidence for supporting the argument on seed transmission of SPLCV in Sweetpotato, the claim that SPLCV is not seed transmitted in sweetpotato that have not been adequately addressed in the manuscript.

Major concern:

To determine whether SPLCV is transmitted via seeds in Sweetpotato, the published literature (Kim et al., 2015) firstly indicates that SPLCV can be transmitted via seeds in sweet potato plants with up to 15% transmission level, all the sequence of a spanning region in the SPLCV genome from leaf, petal, floral tissues, seeds and seedlings were identical in sequence to that in the Korean isolate of SPLCV. Most importantly, SPLCV was also detected in whole dry seeds, seed coats with part of the endosperm, and internal seed tissue consisting of embryo and endosperm, and the accumulation of SPLCV level was detected by PCR and Southern blot hybridization in developing leaves. Therefore, the authors need to present some comparative evidence to support their argument again the previous claim in the manuscript.

Response:  

It is important to distinguish between ‘seed-borne’ and ‘seed transmission’ when describing the biological properties of a plant viruses. Seed transmission of a virus can have a significant impact not only on disease epidemiology and management but also on international seed trade, whereas the seed-borne nature of a virus is less concerning. In the present study, we detected the presence of SPLCV on sweetpotato whole seeds and seed coats (testa) as well as other vegetative tissues through real-time PCR detection, confirming SPLCV as seed-borne (contaminant) on sweetpotato [22]. Seed transmission indicates that virus particles presented on/in the seed could be transmissible to the offspring, resulting in a new infection in the geminated seedlings. Instead of using artificial separation of different seed parts (seed coats with part of the endosperm or an internal seed tissue of embryo and endosperm) as in Kim et al. 2015 [22] (which might result in a potential cross-contamination of seed parts when testing by PCR), we used germinated seedlings in petri dishes through natural separation of embryos from seed coats. SPLCV was detected on the seed coats but not in the embryos (Table 5), suggesting the virus is not seed transmissible. This distinction is now incorporated into the discussion.

More importantly, large scale seedling grow-out experiments over 4 years (23,034 sweetpotato seedlings of 118 genotype accessions) in the present study did not support the true seed transmission. Previous study used only approximately 100 germinated seedlings and found 15% seed transmission of SPLCV to seedlings (Kim et al. 2015). Although we used different genotypes of sweetpotato as those in the previous study, Kim et al. (2015) germinated seedlings in the greenhouse first and then moved to a grow chamber, which may provide an opportunity for accidental whitefly transmission before seedlings were moved to the growth chamber. Even in some of our grow-out experiments when seedlings were germinated on open benches in an environment-controlled greenhouse, we detected some small number of seedlings positive for SPLCV, which could not be supported when seedlings were germinated and grown under more stringent insect-proof conditions (additional insect-proof nettings in a greenhouse or in growth chambers). 

Furthermore, there is no evidence of seed transmission of tomato yellow leaf curl virus (TYLCV), another begomovirus in tomato and Nicotiana benthamiana (Rosas-Díaz, et al., 2017; Pérez-Padilla, et al., 2020), even though the initial study from the same group in Korea observed a high level of seed transmission of TYLCV on tomato (Kil et al., 2016).

Round 2

Reviewer 2 Report

This revised version of the manuscript improved in terms of addressed my prior comments, I will be happy to recommend it for acceptance in light of changes made in response to the journal formatting guidelines.